# Gut microbiota confers host resistance to obesity by metabolizing dietary polyunsaturated fatty acids

Junki Miyamoto[1,2], Miki Igarashi[1], Keita Watanabe[1], Shin-ichiro Karaki[3], Hiromi Mukouyama[1], Shigenobu Kishino [4], Xuan Li[1], Atsuhiko Ichimura [5,6], Junichiro Irie [2,7], Yukihiko Sugimoto [2,8], Tetsuya Mizutani[9], Tatsuya Sugawara [10], Takashi Miki [11], Jun Ogawa[4], Daniel J. Drucker[12], Makoto Arita[13,14], Hiroshi Itoh[2,7] & Ikuo Kimura [1,2]

Gut microbiota mediates the effects of diet, thereby modifying host metabolism and the incidence of metabolic disorders. Increased consumption of omega-6 polyunsaturated fatty acid (PUFA) that is abundant in Western diet contributes to obesity and related diseases. Although gut-microbiota-related metabolic pathways of dietary PUFAs were recently elucidated, the effects on host physiological function remain unclear. Here, we demonstrate that gut microbiota confers host resistance to high-fat diet (HFD)-induced obesity by modulating dietary PUFAs metabolism. Supplementation of 10-hydroxy-*cis*-12-octadecenoic acid (HYA), an initial linoleic acid-related gut-microbial metabolite, attenuates HFD-induced obesity in mice without eliciting arachidonic acid-mediated adipose inflammation and by improving metabolic condition via free fatty acid receptors. Moreover, *Lactobacillus*-colonized mice show similar effects with elevated HYA levels. Our findings illustrate the interplay between gut microbiota and host energy metabolism via the metabolites of dietary omega-6-FAs thereby shedding light on the prevention and treatment of metabolic disorders by targeting gut microbial metabolites.

[1] Department of Applied Biological Science, Graduate School of Agriculture, Tokyo University of Agriculture and Technology, Fuchu-shi, Tokyo 183-8509, Japan. [2] AMED-CREST, Japan Agency for Medical Research and Development, Chiyoda-ku, Tokyo 100-0004, Japan. [3] Laboratory of Physiology, Department of Environmental and Life Sciences, University of Shizuoka, Shizuoka 422-8526, Japan. [4] Division of Applied Life Sciences, Graduate School of Agriculture, Kyoto University, Kitashirakawa Oiwake-cho, Sakyo-ku, Kyoto 606-8502, Japan. [5] Department of Biological Chemistry, Graduate School of Pharmaceutical Sciences, Kyoto University, Kyoto 606-8501, Japan. [6] K-CONNEX, Keihanshin Consortium for Fostering the Next Generation of Global Leaders in Research, Kyoto 606-8501, Japan. [7] Department of Endocrinology, Metabolism and Nephrology, School of Medicine, Keio University, Shinjuku-ku, Tokyo 160-8582, Japan. [8] Department of Pharmaceutical Biochemistry, Graduate School of Pharmaceutical Sciences, Kumamoto University, Kumamoto 862-0973, Japan. [9] Research and Education Center for Prevention of Global Infectious Diseases of Animals, Faculty of Agriculture, Tokyo University of Agriculture and Technology, Tokyo 183-0045, Japan. [10] Division of Applied Biosciences, Graduate School of Agriculture, Kyoto University, Kyoto 606-8502, Japan. [11] Department of Medical Physiology, Graduate School of Medicine, Chiba University, Chiba 260-8670, Japan. [12] Department of Medicine, Lunenfeld Tanenbaum Research Institute, Mount Sinai Hospital, University of Toronto, Toronto ON M5S 2J7, Canada. [13] Division of Physiological Chemistry and Metabolism, Faculty of Pharmacy, Keio University, 1-5-30, Shibakoen, Minato-ku, Tokyo 105-0011, Japan. [14] Laboratory for Metabolomics, RIKEN Center for Integrative Medical Sciences, Kanagawa 230-0045, Japan. Correspondence and requests for materials should be addressed to I.K. (email: ikimura@cc.tuat.ac.jp)

D iet is the most important factor affecting host nutrition and metabolism; however, dysregulation of energy homeostasis due to excess food intake leads to obesity. Obesity progression occurs due to a long-term imbalance between energy intake and expenditure that influences multiple pathways via metabolites and hormones. Excess food intake, especially high-calorie diets, such as high-fat and high-sugar diets, is considered the greatest risk factor in the development of obesity[1]. Especially, high-fat diet (HFD)-induced obesity results in changes in gut-microbial composition, as well as reductions in microbial diversity and changes in specific bacterial taxa[2–5], despite gut microbiota is a key interface for energy acquisition because it can convert dietary substrates to host nutrition in a composition-dependent manner[2,6].

During the previous three decades, total fat and saturated-fat intake as a percentage of total calories continued to decrease in Western diets[7,8]. However, the intake of omega-6 fat that includes primarily the essential fatty acid (FA) linoleic acid (LA), has increased, whereas intake of the omega-3 fat that includes other essential FAs alpha-linolenic acid (αLA) has decreased, thereby significantly increasing the omega-6/omega-3 ratio over 20:1[9]. This change in the composition of polyunsaturated FAs (PUFAs) substantially increases the incidence and prevalence of overweight and obesity[7]. Regarding the physiological effects of these FAs, recent studies demonstrate that PUFA-related metabolic effects not only promote the synthesis of eicosanoids, which act as lipid mediators and are utilized as an energy source, but also affect the free-FA-specific receptors free fatty acid receptor (FFAR)1 and FFAR4 [also known as G-protein-coupled receptor (GPCR) GPR40 and GPR120, respectively][10].

GPR40 is a receptor for medium- and long-chain FAs (LCFAs) that include PUFAs, whereas GPR120 is a receptor exclusively for LCFAs. These receptors couple with the Gq family of G proteins, leading to elevated levels of intracellular calcium ($[Ca^{2+}]i$)[11]. GPR40 is highly expressed in pancreatic insulin-producing β cells, and LCFAs strongly promote glucose-stimulated insulin secretion via GPR40[12]. Dysfunction of GPR120, which is expressed in adipose tissues, leads to obesity, resulting in glucose intolerance and fatty liver accompanied by decreased adipocyte differentiation and lipogenesis, as well as enhanced hepatic lipogenesis[13]. Moreover, GPR120 activation by docosahexaenoic acid exerts anti-inflammatory effects in macrophages[14]. Furthermore, both receptors are expressed in enteroendocrine L cells, and LCFAs promote the secretion of gut hormone, glucagon-like peptide-1 (GLP-1) via these receptors[15].

Gut microbiota mediate saturation of PUFAs derived from dietary fat as a detoxifying mechanism in the gastrointestinal tract[16]. To date, various bacteria, including gut microbes, have been identified as producing PUFA-derived intermediate metabolites[17–20]. Furthermore, in vitro stimulation by and in vivo administration of PUFA-derived bacterial intermediate metabolites result in anti-obesity and anti-inflammatory effects[21–24]; however, the relationship between host influence, the physiological functions of these intermediate metabolites, and gut microbiota remains unclear. In this study, we show that gut microbiota confers host resistance to HFD-induced obesity through the production of PUFA metabolites by investigating the integrated role of gut-microbial PUFA metabolites in host energy regulation in mice.

## Results

### Dietary PUFA-derived gut microbial metabolites.
Microbes transform PUFAs into a variety of FAs, such as LA-derived metabolites (Supplementary Fig. 1a)[17] and αLA-derived metabolites (Supplementary Fig. 1b)[18]. We first examined the gut-microbial PUFA-metabolite profiles in the cecum of normal chow (NC)-fed and HFD-fed mice, finding that LA-derived metabolites existed at predominantly higher levels as compared with those associated with αLA-derived metabolites (Supplementary Fig. 1c), with 10-hydroxy-cis-12-octadecenoic acid (HYA), 10-hydroxyoctadecanoic acid (HYB), 10-hydroxy-trans-11-octadecenoic acid (HYC), 10-oxo-cis-12-octadecenoic acid (KetoA), 10-oxo-octadecanoic acid (KetoB), and 10-oxo-trans-11-octadecenoic acid (KetoC) in LA-derived metabolites specifically attenuated by HFD-feeding for 2 weeks (Fig. 1a). Moreover, by comparing these LA-derived metabolites (HYA, HYB, HYC, KetoA, KetoB, and KetoC) in the cecum of NC- and HFD-fed mice, we observed a significantly reduced level of these metabolites in mice fed a HFD for 2 weeks (Fig. 1b). However, HFD supplemented with LA increased cecal levels of HYA (the initial gut-microbial PUFA-metabolite derived from LA) and KetoA, whereas levels of all metabolites were markedly decreased in both NC- and HFD-fed germ-free (GF) mice (Fig. 1b). Moreover, 16 S rRNA gene amplicon sequencing confirmed that HFD supplemented with LA altered the relative abundance of the major phyla constituting the gut microbiota (Fig. 1c). Specifically, in agreement with previous studies[6], the abundance of Firmicutes markedly increased and that of Bacteroidetes markedly decreased in HFD-fed mice; however, the abundance of Firmicutes was decreased by LA supplementation (Fig. 1c). Additionally, we confirmed that HFD feeding altered gut microbiota composition, as indicated by principal coordinate analysis (PCoA) based on taxonomic datasets (Fig. 1d). Moreover, hierarchical clustering of individual families confirmed the effect of HFD and LA-supplemented HFD on the gut microbiome (Fig. 1e). Although HFD feeding was associated with a drastically decreased abundance of the Lactobacillaceae family belonging to Firmicutes, interestingly, LA supplementation contributed to an increase in the abundance of Lactobacillaceae (Fig. 1e) and a significant expansion of the Lactobacillus genus (Fig. 2a). Moreover, we examined what species were related to HYA production in the Lactobacillus genus by in vitro bacterial-culture screening. Lactobacillus salivarius and Lactobacillus gasseri efficiently produced HYA from LA in 22 Lactobacillus strains, whereas Lactobacillus acidophilus and Lactobacillus johnsonii produced very little HYA (Supplementary Table 1). Similarly, L. salivarius and L. gasseri, as HYA-producing bacteria, were markedly decreased in HFD-fed mice and significantly increased following LA supplementation, whereas L. acidophilus and L. johnsonii, as HYA non-producing bacteria, did not change following LA supplementation (Fig. 2b). Furthermore, we observed suppressed expression of Cla-hy, a key LA-metabolizing enzyme characterized in gut microbes (Supplementary Fig. 1a), in HFD-fed mice and that this suppression was alleviated by LA supplementation, whereas levels of Cla-dh and Cla-er (Supplementary Fig. 1a) did not change following LA supplementation (Fig. 2c). These findings collectively demonstrated that HFD feeding altered gut microbial composition and inhibited the production of PUFA metabolites by the gut microbes; however, PUFA supplementation restored the abundance of the Lactobacillus and the gut microbial PUFA-metabolite HYA.

### Microbial PUFA metabolites improve host metabolic condition.
We then examined the effects of gut-microbial PUFA metabolites on host energy regulation in a mouse model of HFD-induced obesity. HYA is an initial product of gut-microbial metabolites from dietary LA (Supplementary Fig. 1a). In this experiment, 4-week-old mice were fed HFD (control) or HFD supplemented with either 1% HYA as a gut-microbial metabolite or 1% LA, a precursor of HYA for 12 weeks (Supplementary

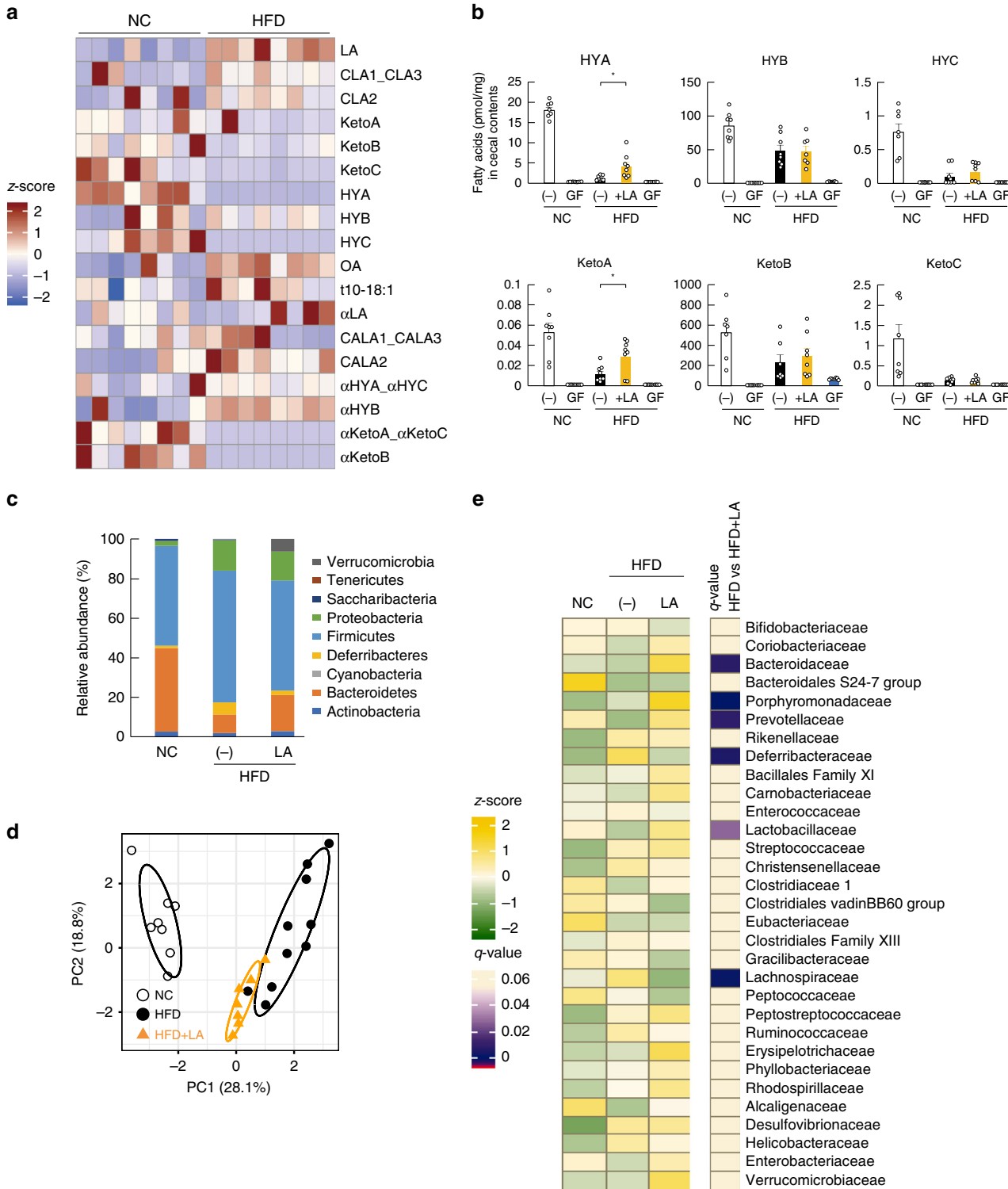

**Fig. 1** Effects of dietary PUFAs on gut microbiota composition and PUFA metabolites. **a** Heat map of gut microbial PUFA metabolites in cecal content ($n = 8$ per group). **b** LA-derived gut microbial PUFA metabolites in cecal content were quantified ($n = 7, 10, 8, 8,$ and 8 per group for HYA; $n = 8, 10, 8, 8,$ and 8 per group for HYB; $n = 7, 10, 8, 8,$ and 10 per group for HYC; $n = 8, 10, 8, 8,$ and 10 per group for KetoA; $n = 7, 10, 7, 8,$ and 8 per group for KetoB; $n = 8, 9,$ 8, 8, and 8 per group for KetoC). *$P < 0.05$ (Tukey–Kramer test). **c–e** Gut microbial composition was evaluated in order to determine the relative abundance of microbial taxa (**c**), diversity (**d**), and abundance of the bacterial domain at the family level (**e**) ($n = 8, 10,$ and 7 per group). (−)_NC represents normal chow-fed mice, and (−)_HFD represents high-fat diet-fed mice. $q < 0.05$. Results are presented as means ± SE. Source data are provided as a Source Data file 1

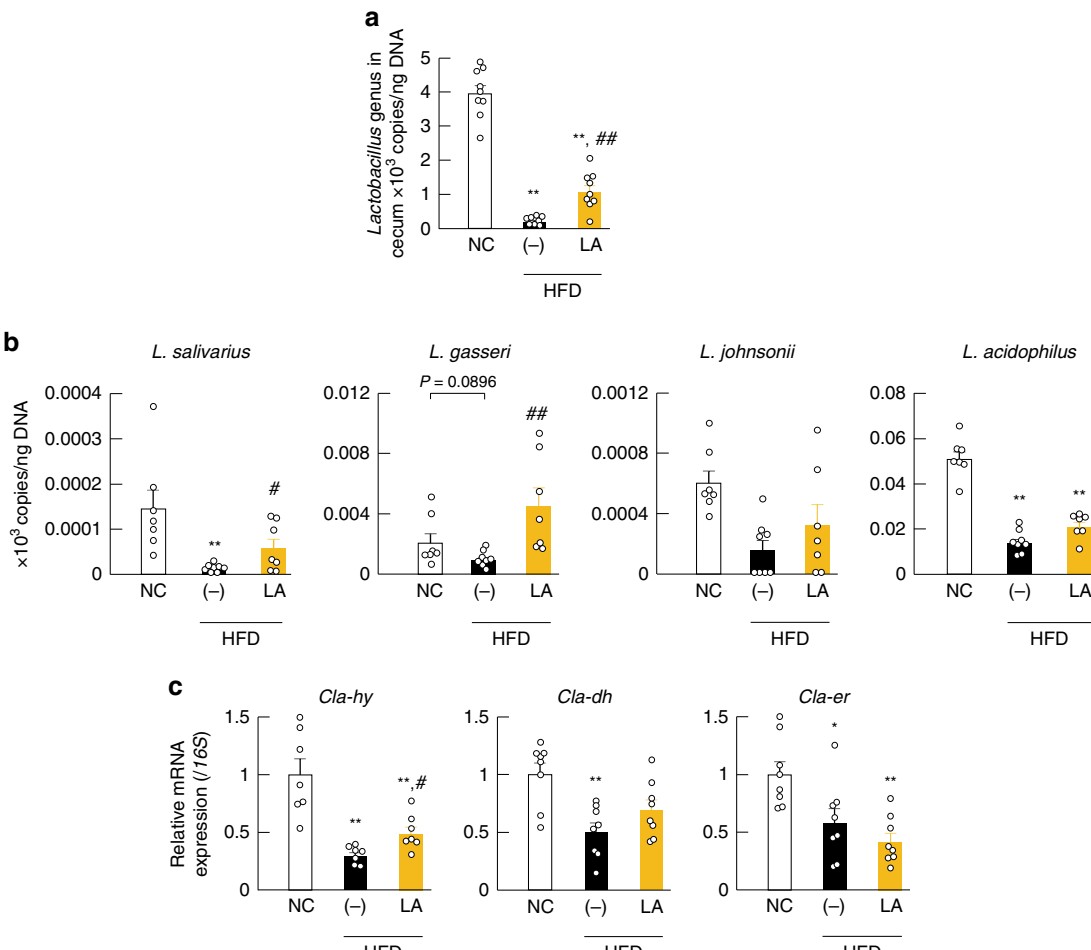

**Fig. 2** HYA-producing lactic acid bacteria. **a** The abundance of *Lactobacillus* was analyzed by qPCR ($n = 9$, 8, and 9 per group). **b** The abundance of the *Lactobacillus* strains ($n = 7$, 8, and 7 per group) and **c** the relative mRNA expression of metabolite-synthesizing enzymes (*Cla-hy*, *Cla-dh*, and *Cla-er*; see Supplementary Fig. 1) ($n = 7$ per group for *Cla-hy*; $n = 8$ per group for *Cla-dh* and *Cla-er*) in cecum were analyzed by qPCR and qRT-PCR. \*\**P* < 0.01, \**P* < 0.05 vs. NC (Tukey–Kramer test). ##*P* < 0.01; #*P* < 0.05 vs. (−) (Tukey–Kramer test). (−)_HFD represents high-fat diet-fed mice. Results are presented as means ± SE. Source data are provided as a Source Data file 2

Table 2). We found that the body weight of HYA-fed mice was significantly lower as compared with that of control and LA-fed mice during growth (Fig. 3a). In addition, the fat mass of white adipose tissue (WAT) was significantly lower in HYA-fed mice than in control and LA-fed mice at 16 weeks of age (Fig. 3b). Moreover, we found a significant decrease in WAT adipocyte size in HYA-fed mice (Fig. 3c), and the food intake of HYA- and LA-fed mice was significantly lower than that of control mice (Fig. 3d). Furthermore, HFD-induced insulin resistance and impaired glucose tolerance, as determined by the insulin tolerance test (ITT) and glucose tolerance test (GTT), respectively, were significantly attenuated in HYA-fed mice as compared with control and LA-fed mice (Fig. 3e, f). Moreover, the plasma glucose and total cholesterol levels of HYA-fed mice were significantly lower than those of control mice (Fig. 3g, h), whereas triglyceride levels were similar among control, HYA-, and LA-fed mice (Fig. 3i). Additionally, we found that levels of the plasma peptide YY (PYY) were significantly higher in HYA-fed mice than in control mice (Fig. 3j), and similarly, that plasma GLP-1 levels were markedly higher in HYA-fed mice than in control and LA-fed mice (Fig. 3k), whereas the plasma insulin levels of HYA-fed mice were significantly lower than those of control mice (Fig. 3l). Furthermore, we observed a significant increase in the level of fecal triglycerides of HYA-fed mice (Fig. 3m). These findings

suggested that HYA supplementation suppressed appetite and improved metabolic condition, thereby inducing greater resistance to HFD-induced obesity, even in the presence of PUFAs. In addition, we performed the same experiment under 0.5% HYA-supplemented HFD-fed conditions equivalent to physiologically relevant concentrations of HYA in the cecum of NC-fed mice (Supplementary Fig. 2a). The results showed that 0.5% HYA supplementation suppressed appetite and improved metabolic condition, thereby inducing greater resistance to HFD-induced obesity, similar to 1% HYA supplementation (Supplementary Fig. 2b–m).

**Microbial PUFA metabolites and adipose inflammatory response**. We also examined the influence of HYA- or LA-feeding on host lipid metabolism over a period of 12 weeks. Interestingly, long-term HFD-feeding for 12 weeks compared with NC feeding resulted in elevated KetoB and HYB levels in the cecum relative to levels observed resulting from HFD-feeding for 2 weeks, whereas long-term HFD-feeding for 12 weeks, similar to HFD-feeding for 2 weeks, resulted in decreased HYA levels (Supplementary Fig. 3a). In addition, long-term HFD-feeding for 12 weeks, similar to HFD-feeding for 2 weeks, suppressed *Cla-hy* expression, whereas long-term HFD-feeding for 12 weeks increased *Cla-dh* and *Cla-er*

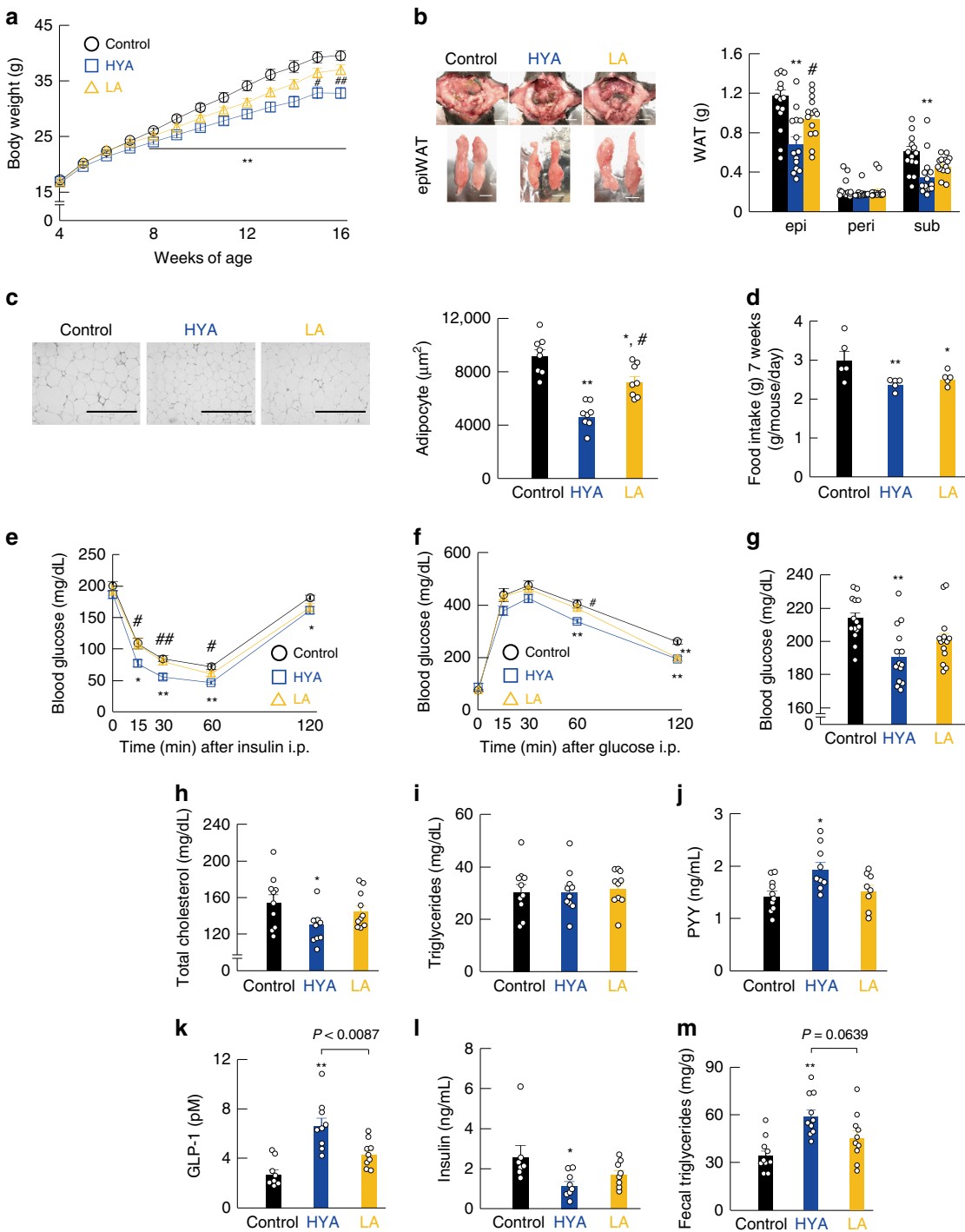

**Fig. 3** Gut microbial PUFA metabolites improve host metabolic conditions. Changes in **a** body weight and **b** representative macroscopic appearance and tissue weights ($n = 14$ per group). Scale bar; 1 cm. epi, epididymal; peri, perirenal; sub, subcutaneous. **c** Hematoxylin–eosin (H&E)–stained WAT and the mean area of adipocytes ($n = 8$ per group). Scale bar, 400 μm. **d** Daily food intake measured at 7 weeks of age ($n = 5$ per group). **e** ITT ($n = 10$ per group) and **f** GTT ($n = 10$ per group) were analyzed at 13–14 weeks of age. **g** Blood glucose ($n = 14$ per group), **h** total plasma cholesterol ($n = 10$, 9, and 10 per group), **i** triglyceride ($n = 10$ per group), **j** PYY ($n = 10$, 9, and 8 per group), **k** GLP-1 ($n = 8$, 9, and 9 per group), and **l** insulin ($n = 7$, 8, and 8 per group) levels were measured at the end of the experimental period. **m** Fecal triglyceride levels were measured at 16 weeks of age ($n = 10$ per group). $**P < 0.01$; $*P < 0.05$ vs. control (Tukey–Kramer test). $##P < 0.01$; $#P < 0.05$ vs. HYA (Tukey–Kramer test). Results are presented as means ± SE. Source data are provided as a Source Data file 3

expression as compared with HFD-feeding for 2 weeks (Supplementary Fig. 3b). Examination of the 16S gene-base cecal microbiome showed that long-term HFD-feeding, similar to LA-supplemented HFD feeding for 2 weeks, increased the abundance of the Lactobacillaceae family relative to that in controls (Supplementary Fig. 3c). Interestingly, we found that HYA supplementation increased Lactobacillaceae abundance to a greater extent than LA supplementation (Supplementary Fig. 3c), with hierarchical clustering of individual families confirming the effect of HFD, LA-supplemented HFD, and HYA-supplemented HFD

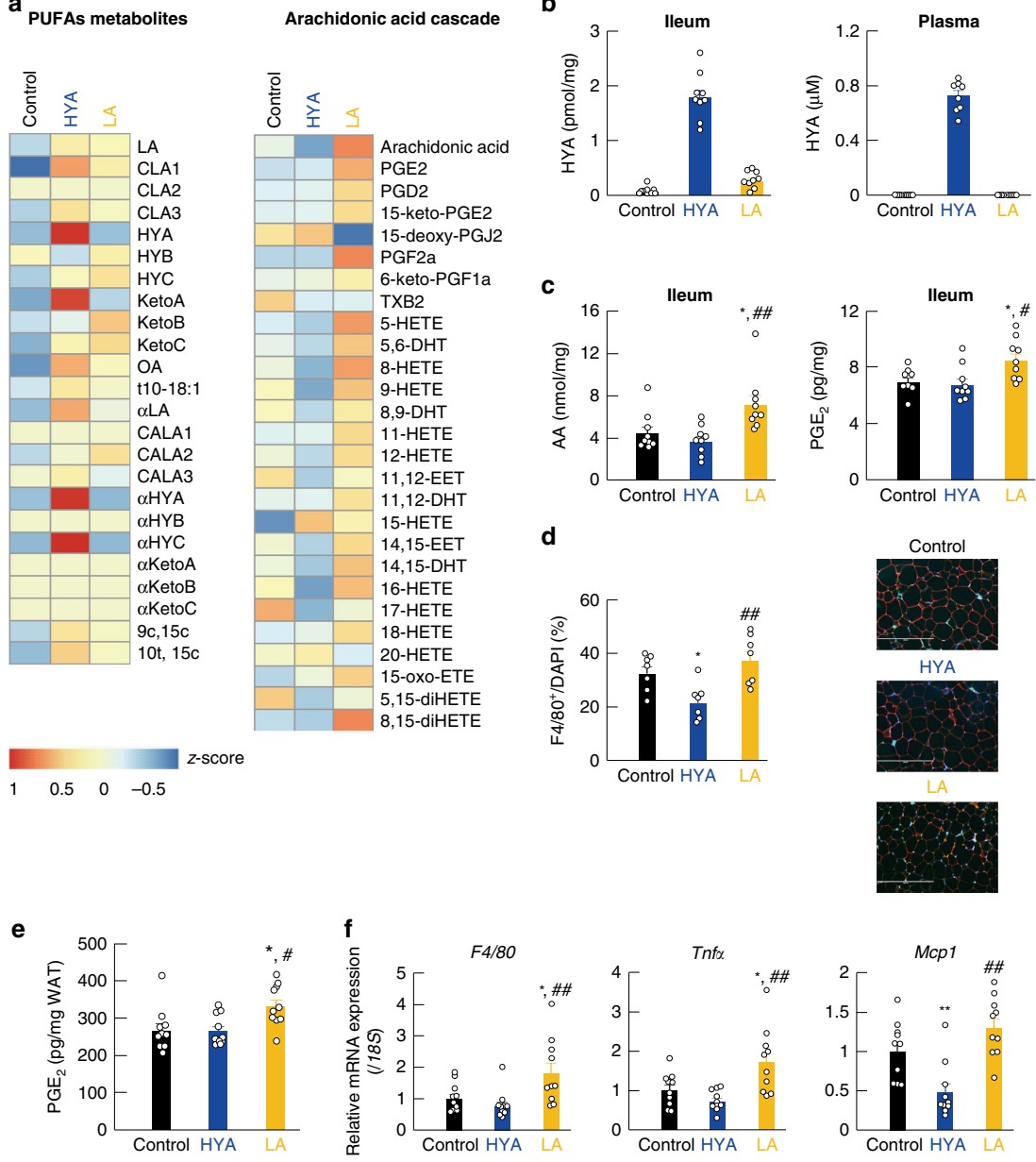

**Fig. 4** Gut microbial PUFA metabolites and adipose inflammatory response. **a** Heat map of FA profiles in the ileum ($n = 4$ per group). **b** HYA was detected in the ileum (left) and plasma (right) ($n = 9$ per group in the ileum; $n = 9$, 8, and 9 per group in the plasma). **c** Arachidonic acid (left) and PGE2 (right) were quantified in the ileum ($n = 9$ per group for arachidonic acid; $n = 8$, 9, and 9 per group for PGE2). **d** WAT sections were labeled by F4/80 (green), caveolin-1 (red), and DAPI (blue), and F4/80-positive cells were measured ($n = 7$ per group). Scale bar, 400 μm. **e** Levels of adipose PGE2 ($n = 10$ per group). **f** mRNA expression of F4/80, Tnfα, and Mcp1 in the WAT of HFD-induced obese mice ($n = 10$ per group). $**P < 0.01$; $*P < 0.05$ vs. control (Tukey–Kramer test). $^{\#\#}P < 0.01$; $^{\#}P < 0.05$ vs. HYA (Tukey–Kramer test). Results are presented as means ± SE. Source data are provided as a Source Data file 4

feeding on the gut microbiome (Supplementary Fig. 3d). LCFAs, such as PUFAs, are mainly absorbed in the small intestine; therefore, we next examined the tissue-transferred FA profile in the ileum by lipid-metabolome analysis, finding that HYA supplementation increased KetoA levels in LA-derived gut microbial metabolites in the ileum (Fig. 4a) and significantly increased HYA levels in ileum and plasma (Fig. 4b). Subsequently, we found that LA-supplementation increased the abundance of FA metabolites related to the arachidonic acid (AA) cascade as compared with that observed in control mice, although levels of these FA metabolites in HYA-fed mice were similar to those in control mice or slightly decreased (Fig. 4a). Moreover, based on quantitative analysis in the ileum, AA and prostaglandin E2 (PGE2)

levels in LA-fed mice were significantly increased as compared with those in control mice, whereas levels in HYA-fed mice were similar to those in control mice (Fig. 4c). Since prostaglandins and thromboxane via the AA cascade are considered to be lipid mediators of the inflammatory response[25], we examined whether LA supplementation augmented the adipose inflammation response via the AA cascade. As anticipated, we found a significant increase in F4/80-positive macrophages in the WAT of LA-fed mice as compared with that of HYA-fed mice (Fig. 4d). In addition, adipose PGE2 levels in LA-fed mice were significantly higher than those in control mice, whereas comparable levels were observed between control and HYA-fed mice (Fig. 4e). Furthermore, we found that the expression of F4/80, the

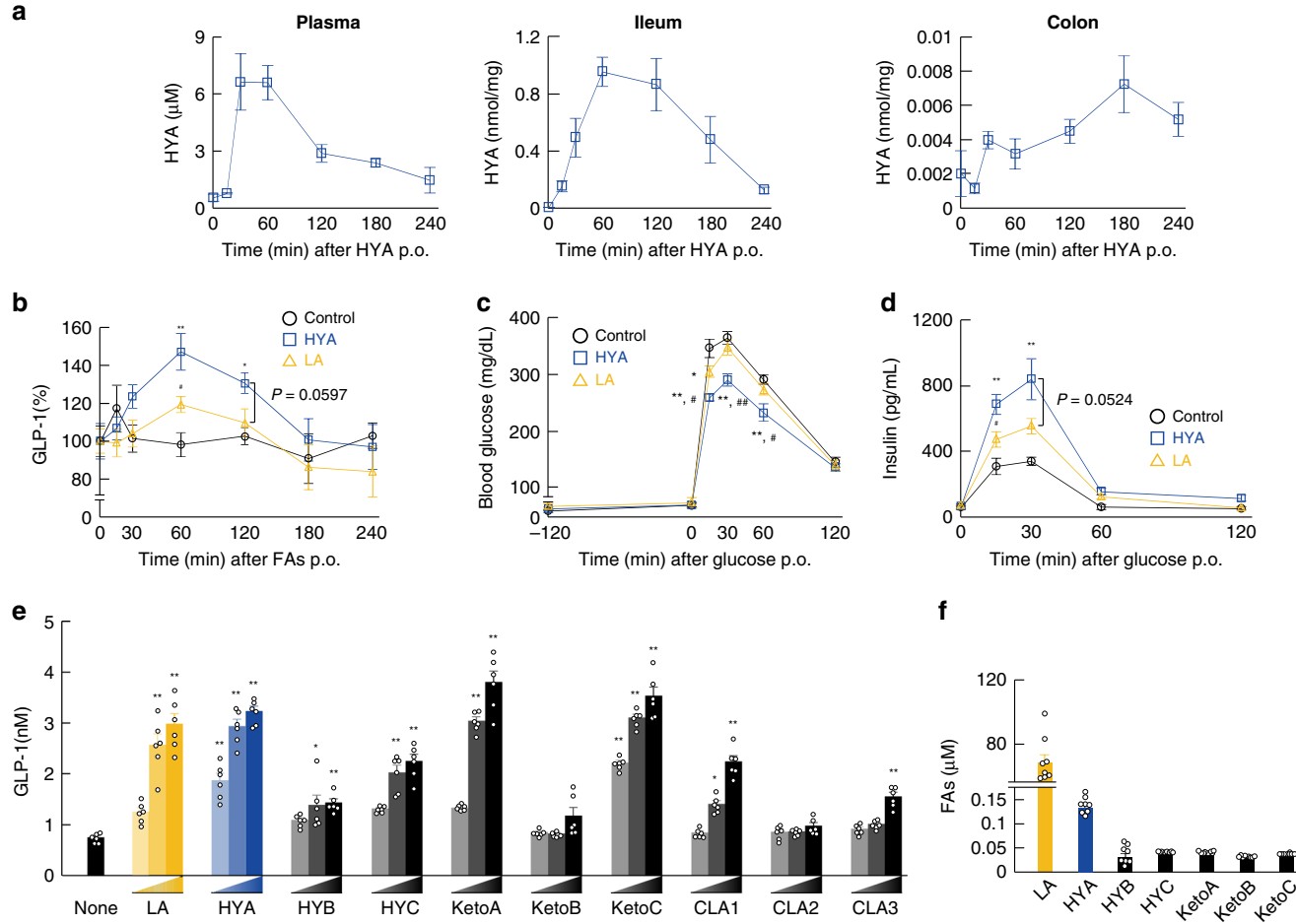

**Fig. 5** HYA directly regulates glucose homeostasis and GLP-1 release. Individual FAs (HYA and LA; 1 g/kg) were administered by gavage, followed by **a** HYA quantification in plasma (left), ileum (center), and colon (right) ($n = 8$ animals). **b** Time-course changes in plasma GLP-1 from the tail vein was measured after oral administration of FAs ($n = 7$ animals per group). Basal GLP-1 concentration at time 0 was set as 100%. **c** OGTT was analyzed 2 h after individual FA administration (HYA and LA; 1 g/kg) by gavage ($n = 8$ animals per group). **d** Individual FAs were administered, and 2 h later, time-course changes in plasma insulin from the tail vein were measured after oral administration of glucose ($n = 8$ animals per group). $**P < 0.01$; $*P < 0.05$ vs. control (Tukey–Kramer test). $\#\#P < 0.01$; $\#P < 0.05$ vs. HYA (Tukey–Kramer test). Results are presented as means ± SE. **e** STC-1 cells were treated with LA-derived gut microbial metabolites [in a dose-dependent manner (20, 100, and 200 μM)], and GLP-1 concentration was measured in culture medium ($n = 6$ independent cultures from three biological replicates). $**P < 0.01$; $*P < 0.05$ vs. None (Tukey–Kramer test). **f** LA-derived gut microbial metabolites were detected in plasma of NC-fed mice ($n = 8$ animals). Results are presented as means ± SE. Source data are provided as a Source Data file 5

inflammatory marker tumor necrosis factor α (TNF-α), and monocyte chemoattractant protein-1 (MCP-1; also known as CCL2) was markedly increased in LA-fed mice as compared with control and HYA-fed mice (Fig. 4f). Therefore, in contrast to HYA supplementation, LA supplementation promoted the progression of adipose inflammation via the AA cascade.

**HYA directly regulates glucose homeostasis and GLP-1 release.** We then investigated the effects of the gut microbial PUFA-metabolite HYA on GLP-1 secretion and glucose homeostasis. The incretin GLP-1, gut hormone that stimulates glucose-induced insulin secretion and inhibits food intake, is secreted from enteroendocrine L cells, which are primarily found in the ileum and colon[26]. Following oral administration of HYA (1 g/kg), HYA concentrations peaked 1 h after administration in plasma and the ileum, and 3 h after administration in the colon (Fig. 5a). In addition, HYA administration increased gut microbial HYA metabolites, such as HYC, KetoA, and KetoC, as well as HYA concentration in the ileum (Supplementary Fig. 4a). Similar to the peak time in the ileum, we found that plasma GLP-1 levels

following both HYA and LA administration also peaked 1 h after administration, with the peak plasma GLP-1 levels following HYA administration higher than those in controls and LA-administered mice from 0.5 h to 3 h (Fig. 5b). Therefore, we performed oral GTT (OGTT) and intraperitoneal GTT (IPGTT) based on increases in GLP-1 levels following HYA administration at 1 h. Following oral administration of LA or HYA and after oral or intraperitoneal administration of glucose 2-h later, we found that HYA administration significantly suppressed increases in blood glucose as compared with that found in control and LA-administered mice (Fig. 5c and Supplementary Fig. 4b). Moreover, plasma insulin levels following glucose administration peaked at 30 min, and glucose-induced insulin secretion levels in HYA-administered mice were higher than those in control and LA-administered mice (Fig. 5d). Next, we examined whether HYA induces GLP-1 secretion using the mouse intestinal secretin tumor-cell line STC-1. We found that the LA-derived gut microbial PUFA metabolites KetoA, KetoC, and HYA, in particular, strongly induced GLP-1 secretion in a dose-dependent manner in STC-1 cells, whereas HYB and KetoB hardly induced these effects (Fig. 5e). In addition, LA-derived gut microbial

PUFA metabolites including HYA, exhibited similar effects in GLUTag cells, an intestinal murine L-cell line (Supplementary Fig. 4c). Under physiological conditions, although HYA levels were the highest among HYA, HYC, KetoA, and KetoC levels in plasma, we found that the levels of all of them were markedly lower than those of LA in plasma (Fig. 5f). Our findings indicated that acute administration of HYA promoted GLP-1 secretion in the intestinal environment and improved glucose homeostasis.

**HYA contributes to host metabolic condition via GPCRs.** We further investigated HYA-mediated GLP-1 secretion signaling. A previous study reported that PUFAs promote GLP-1 secretion via GPR40 and GPR120 as FFARs[27]. Using a heterologous expression system, we found that HYA, KetoA, and KetoC rather than HYC, HYB, or KetoB strongly increased $[Ca^{2+}]i$ levels as compared to LA in both GPR40- and GPR120-overexpressing HEK293 cells, whereas these effects were not observed in doxycycline (−) control HEK293 cells (Fig. 6a, b). Treatment of STC-1 cells with small-interfering (si)RNAs for GPR40 and GPR120 (Supplementary Fig. 5) significantly inhibited HYA- or LA-induced GLP-1 secretion, whereas HYB exhibited no effects (Fig. 6c). We then examined the intracellular signaling mechanism whereby HYA mediates GLP-1 secretion through GPR40 and GPR120. Because GPR40 and GPR120 couple with Gq, these receptors elevate the level of $[Ca^{2+}]i$ and activate phospholipase C (PLC)[28]. We found that HYA- or LA-mediated GLP-1 secretion was effectively blocked by treatment with the mitogen-activated protein kinase (MAPK) kinase inhibitor U0126 (Fig. 6d) and the PLC inhibitor U73122 (Fig. 6e) but not with the CaM kinase II inhibitor KN-62 (Fig. 6f). Therefore, HYA-mediated GLP-1 secretion was mediated through a GPR40− and GPR120−PLC−MAPK cascade rather than via Ca signaling. Next, we generated GPR40- and GPR120-deficient mice using the CRISPR/Cas9 system in order to examine the effects of HYA via GPR40 and GPR120 in vivo (Supplementary Fig. 6a−c and Supplementary Fig. 7a−c). As expected, we found that the HYA-induced increase in GLP-1 secretion in wild-type mice was abolished in both GPR40- and GPR120-deficient mice (Fig. 6g). Moreover, suppression of the increase in blood glucose level following glucose administration by HYA pre-oral administration in wild-type mice was also abolished in both GPR40- and GPR120-deficient mice (Fig. 6h). Therefore, HYA administration promoted GLP-1 secretion and improved glucose homeostasis via activation of GPR40 and GPR120. In addition, we examined the mechanism of HYA-mediated suppression of lipid absorption in the gut. The PGE2 regulates intestinal peristalsis via the EP3 receptor[29], and administration of an EP3 agonist promotes intestinal peristalsis (Supplementary Fig. 8a). Interestingly, we found that HYA was a low-affinity ligand for EP3, and that the observed effect of its administration was higher as compared with that observed with LA (Supplementary Fig. 8b). Moreover, we found that HYA promoted Gq-coupled EP3-mediated elevation in $[Ca^{2+}]i$ in EP3-overexpressing Chem-1 cells. Furthermore, oral administration of HYA and LA significantly promoted intestinal peristalsis as compared with that observed in controls, with the effect of HYA greater than that of LA (Supplementary Fig. 8c). In the presence of piroxicam, the addition of an EP3 agonist induced frequent contractions that mimicked spontaneous giant contractions (GCs), and we found that the frequency and amplitude of HYA-induced GC-like contractions increased in a dose-dependent manner, with these effects blocked by treatment with an EP3 antagonist (Supplementary Fig. 8d). Our findings indicated that HYA promoted intestinal peristalsis by acting as a low-affinity EP3 agonist to suppress lipid absorption in the gut.

**HYA (+) *Lactobacillus* contributes to host metabolic benefits.** We then performed microbial transplantation experiments using HYA-producing gut microbes in order to clarify whether HYA production by gut microbiota contributes to host metabolic improvement. Based on HYA yield (Supplementary Table 1), we selected *L. salivarius* (JCM1044, JCM1042, and JCM1231) as the HYA (+) strain and *L. johnsonii* (JCM1022 and JCM8791) and *L. acidophilus* (JCM1229) as HYA (−) strains and confirmed their intestinal-bacterial colonization (Supplementary Fig. 9a, b). Five weeks after colonization, we found that the fecal HYA levels in HYA (+)-colonized mice were significantly higher than that in GF mice, with no discernable change in this level observed between groups (Supplementary Fig. 9c). After feeding an HFD to *Lactobacillus*-colonized 4-week-old mice for 12 weeks, we found that body weight during growth and fat mass at 16 weeks of age in HYA (+)-colonized mice significantly decreased as compared with those in HYA (−)-colonized mice (Fig. 7a, b). Similar to our observations in HYA-supplemented HFD-fed mice, plasma glucose levels in HYA (+)-colonized mice were also significantly lower than those in HYA (−)-colonized mice (Fig. 7c), whereas total cholesterol and triglyceride levels were similar between these groups (Fig. 7d, e). Moreover, GLP-1 levels were significantly higher in HYA (+)-colonized mice relative to those in HYA (−)-colonized mice (Fig. 7f), whereas plasma insulin levels in HYA ( + )-colonized mice were significantly lower than those in HYA (−)-colonized mice (Fig. 7g). In addition, we found significantly decreased adipose *Tnfα* expression and increased fecal triglycerides in HYA (+)-colonized mice relative to these levels in HYA (−)-colonized mice (Fig. 7h, i), and that glucose clearance as assessed by OGTT was improved in HYA (+)-colonized mice as compared with HYA (−)-colonized mice (Fig. 7j). Furthermore, cecal HYA levels in HYA (+)-colonized mice were significantly higher than those in GF and HYA (−)-colonized mice and similar to those in conventionally raised mice (Fig. 7k). Therefore, we found that HYA-producing gut microbial Lactobacilli improved metabolic conditions. In addition, although in fecal microbiota-transplantation (FMT) experiments, OGTT was significantly improved in FMT mice from long-term HYA-supplemented HFD-fed mice as compared with FMT mice from control HFD-fed mice (Supplementary Fig. 9d), HYA-supplemented HFD feeding also significantly suppressed weight gain as compared with control HFD feeding during growth under metabolic cage bleeding in order to exclude the influence of coprophagy (Supplementary Fig. 9e). These results indicated that in addition to HYA direct effects, HYA-mediated changes in gut microbial composition also influenced to the host metabolic condition. Collectively, our results indicated that gut microbiota conferred host resistance to obesity by metabolizing dietary PUFAs into HYA, which functioned to improve host metabolic homeostasis.

**Discussion**

Excessive intake of dietary PUFAs, especially omega-6 FAs, such as LAs, and unbalances the omega-6/omega-3 ratio contribute to metabolic disease along with chronic inflammation. Recently, several metabolic pathways of dietary PUFAs by gut microbiota have been identified; however, an integrated understanding of the effects of these PUFA metabolites on host physiological function remains unavailable. In this study, we found that gut microbiota conferred host resistance to HFD-induced obesity through the production of PUFA metabolites (Fig. 8).

Using lipid-metabolome analysis, we found minimal gut-microbial PUFA metabolites from omega-3 FAs under physiological conditions in the cecum of NC-fed mice; however, we found robust levels of omega-6 FA-derived gut-microbial PUFA

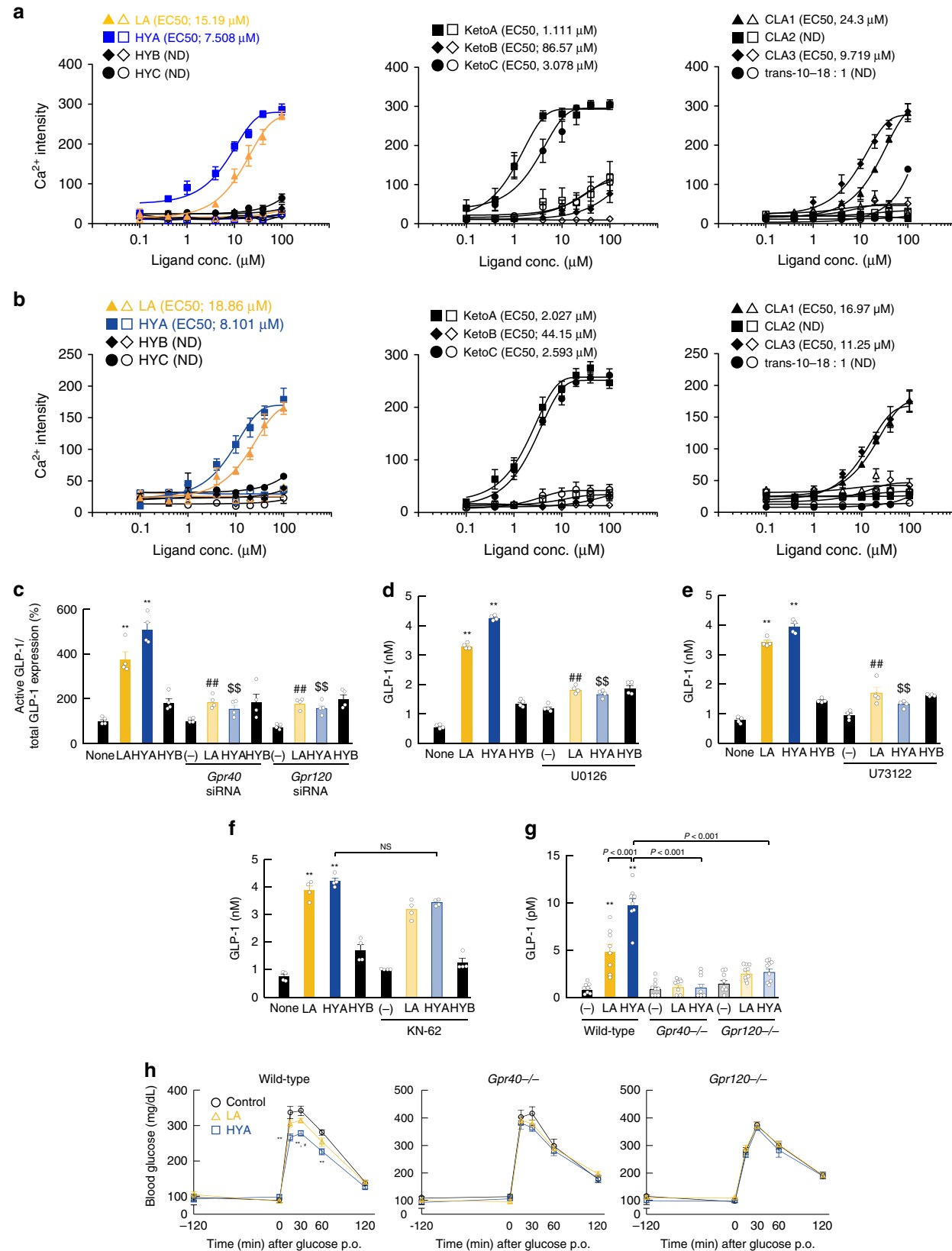

metabolites. Moreover, among all omega-6 FA-derived metabolites, HYA, HYB, HYC, KetoA, KetoB, and KetoC levels derived from LA were markedly decreased under HFD-fed condition for 2 weeks and barely produced under GF conditions. Specifically, long-term HFD feeding for 12 weeks induced substantial increases in KetoB and HYB levels. In addition, we observed that HYB, HYC, and KetoB minimally activated GPR40 and GPR120. Based on these findings, we determined that HYA was not only the initial PUFA-metabolite derived from LA but also the most important gut microbial PUFA-metabolite that influenced host

**Fig. 6** HYA contributes to host metabolic condition via GPR40 and GPR120. Mobilization of [Ca$^{2+}$]i induced by LA-derived gut microbial metabolites was monitored in Flp in **a** hGPR40 or **b** hGPR120 T-REx HEK293 cells. Data are presented as Ca$^{2+}$ intensity. Cells were cultured for 24 h and then treated with or without 10 μg/mL doxycycline ($n = 8$ independent cultures with doxycycline from three biological replicates; $n = 6$ independent cultures without doxycycline from two biological replicates). Closed symbols represent values from cells treated with doxycycline, and open symbols denote untreated groups. **c**–**f** The inhibitory effects of **c** *Gpr40* and *Gpr120* siRNA, **d** MEK inhibitor (U0126), **e** PLC inhibitor (U73122), and **f** CaMKII inhibitor (KN-62) on GLP-1 secretion following LA, HYA, or HYB treatment ($n = 4$ independent cultures from two biological replicates). **$P < 0.01$ vs. None (Tukey–Kramer test). ##$P < 0.01$; #$P < 0.05$ vs. LA (Tukey–Kramer test). $\$\$$$P < 0.01$ vs. HYA (Tukey–Kramer test). (−) represents untreated cells with siRNA or antagonist. Results are presented as means ± SE. **g** GLP-1 concentration and **h** OGTT in wild-type (left, $n = 10$ animals per group), *Gpr40*-deficient (middle, $n = 8$ animals per group), and *Gpr120*-deficient (right, $n = 9$, 10, and 9 animals per group) mice were analyzed 2 h after FA administration. **$P < 0.01$ vs. Control (Tukey–Kramer test). #$P < 0.05$ vs. LA (Tukey–Kramer test). (−) represents the mice without FA administrations. Results are presented as means ± SE. Source data are provided as a Source Data file 6

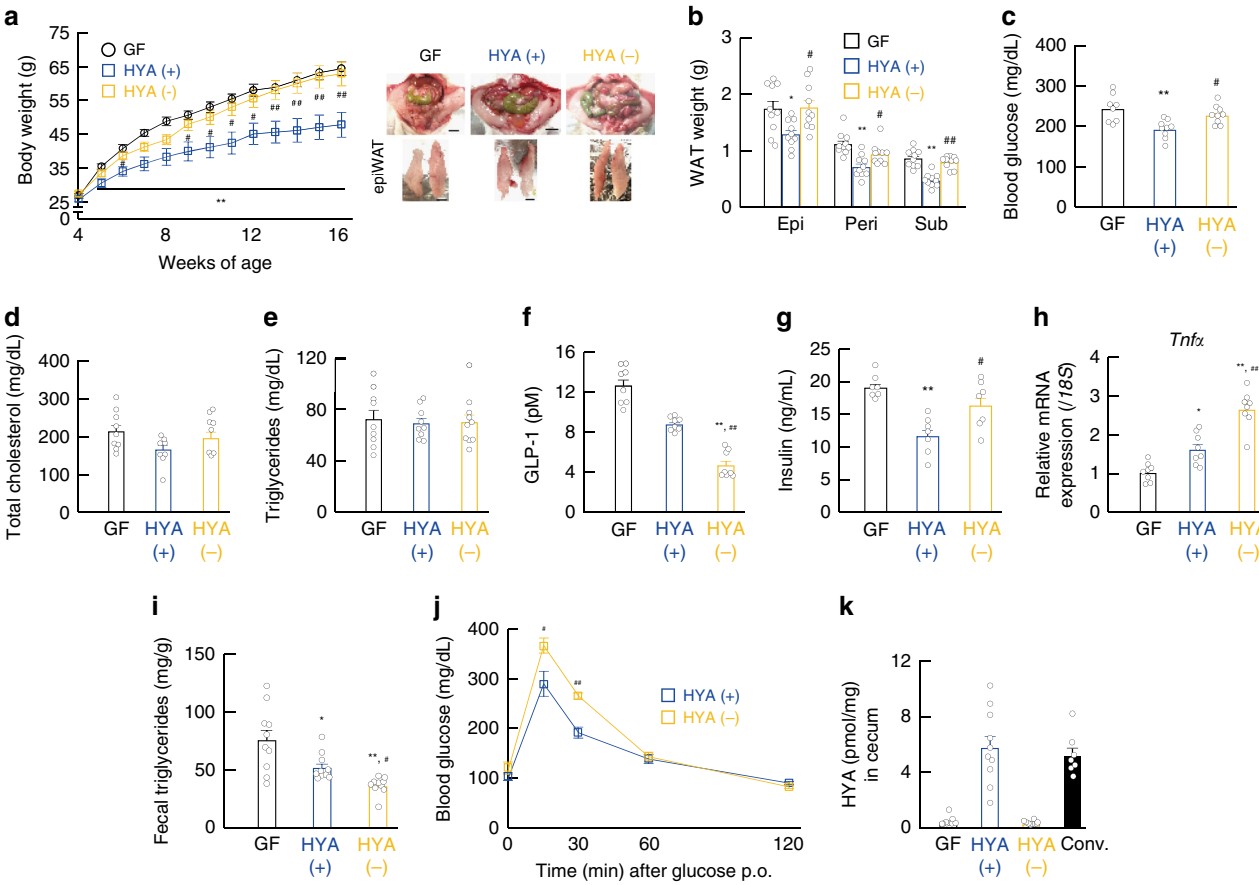

**Fig. 7** HYA-producing *Lactobacillus* contributes to host metabolic improvement. Changes in **a** body weight ($n = 10$, 10, and 8 per group) and **b** tissue weight ($n = 10$ per group) at 16 weeks of age were measured in GF and gnotobiotic mice suffering from HFD-induced obesity. Scale bar; 1 cm. epi, epididymal; peri, perirenal; sub, subcutaneous. **c** Blood glucose ($n = 8$, 9, and 9 per group), **d** total plasma cholesterol ($n = 8$, 7, and 7 per group), **e** triglyceride ($n = 9$, 9, and 10 per group), **f** GLP-1 ($n = 8$ per group), and **g** insulin ($n = 7$ per group) levels were measured at the end of the experimental period. **h** mRNA expression of *Tnfα* in the WAT of HFD-induced obesity ($n = 8$ per group). **i** Fecal triglyceride levels were measured at 16 weeks of age ($n = 10$ per group). **j** After colonization of lactic acid bacteria for 1 week, OGTT was assessed ($n = 7$ per group). **$P < 0.01$; *$P < 0.05$ vs. GF mice (Tukey–Kramer test). ##$P < 0.01$; #$P < 0.05$ vs. HYA ( + ) (Tukey–Kramer test). **k** Cecal HYA was quantified in GF and gnotobiotic mice at 16 weeks of age ($n = 9$, 10, 10, and 7 per group). Results are presented as means ± SE. Source data are provided as a Source Data file 7

metabolism. Moreover, under physiological conditions, plasma HYA levels were ~1/400 (~0.15 μM) of plasma LA levels, whereas levels of other gut microbial PUFA metabolites were markedly lower. Even high-dose (1 g/kg) oral administration of HYA elevated plasma HYA level up to ~6 μM, suggesting that the contribution of circulating HYA to the activation of GPR40 or GPR120 was low. In addition, although cecal HYA levels were also ~1/80 (~20 μM) of cecal LA levels, it might be possible that

local increase of HYA levels around HYA-producing bacteria can activated those receptors in the gut. Therefore, gut microbial PUFA metabolites may exhibit their beneficial metabolic effects via the intestinal environment rather than by exerting systemic effects via circulating plasma such as short-chain FAs, the major gut microbial metabolites.

In addition, we found that mice receiving LA supplementation exhibited adipose inflammation due to the production of an

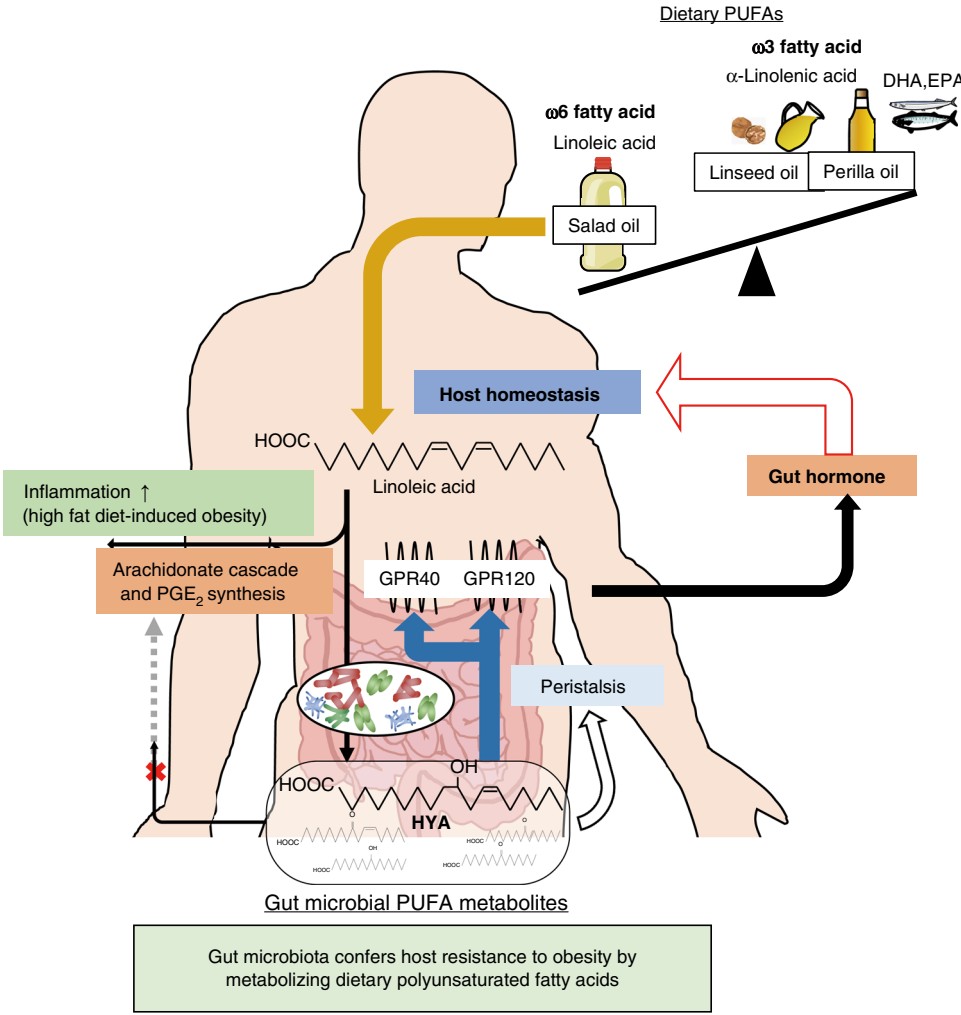

**Fig. 8** Schematic representation. The mechanism by which gut microbial metabolism of dietary PUFAs confers host resistance to obesity

inflammatory lipid mediator via the AA cascade, whereas HYA-supplemented mice did not exhibit these effects. Therefore, our result indicated that gut microbiota suppressed inflammatory responses by converting excessive dietary LA to HYA. LA and HYA supplementation specifically increased the abundance of the Lactobacillaceae in gut microbiota. The genus *Lactobacillus* is facultatively anaerobic and preferentially resides in the small intestine, particularly the ileum, but not the colon, as do most other gut microbes. This accounts for the fact that lipids are absorbed in the small intestine, and increased levels of HYA are found in the ileum and cecum but not the colon. However, LA and HYA supplementation also changed the abundance of some families such as Lachnospiraceae. Therefore, gut microbial PUFA metabolites including HYA, might also exert important host physiological functions via the colon. HYA supplementation and administration not only changed gut microbial composition but also promoted the production of HYA-related gut microbial PUFA metabolites. Moreover, FMT from HYA-supplemented HFD-fed mice improved glucose homeostasis as compared with FMT from HFD-fed control mice. These results indicated that HYA influenced the host metabolic condition not only by HYA itself but also via secondary HYA-mediated changes in the intestinal environment.

Moreover, we found that HYA displayed a high affinity for GPR40 and GPR120 as compared with that of LA, indicating that the gut microbiota locally induced GLP-1 secretion via GPR40 and GPR120 activation in the intestine and that it may be

associated with host glucose homeostasis. Additionally, in vivo acute HYA administration exhibited stronger effects on GLP-1 secretion than in vitro HYA stimulation to STC-1 cells. These results might suggest that further HYA-specific metabolites originating from gut microbiota promote strong GLP-1 secretion. Recently, in addition to HYA, we identified other gut microbial PUFA metabolites with high affinities for GPR40 and GPR120[23]. However, since the concentrations of these other gut microbial PUFA metabolites are lower than HYA, its status as a potential activator of GPR40 and GPR120 might be the most important factor in the associated activity of gut microbial PUFA metabolites under physiological conditions.

Acute HYA administration showed promotion of GLP-1 secretion and improvement of glucose homeostasis. However, our results indicated similar glycemic effects in both OGTT and IPGTT, indicating that other physiological functions besides GLP-1 release, such as pancreatic GPR40-mediated insulin secretion, may also be associated with HYA-mediated insulin responses. Moreover, the mechanism underlying the effect of long-term HYA administration involved metabolic improvement via some other factors. Probably, besides GLP-1-associated appetite suppression and improvement of glucose homeostasis, GLP-1-independent anti-inflammatory effects and intestinal peristalsis also influence metabolic improvements and improvement of glucose homeostasis as secondary effects. Further, other physiological functions via intestinal GPR40 and GPR120, and pancreatic GPR40 may also partially influence host metabolic

effects. Nevertheless, from the viewpoint of gut microbial metabolites, the influence of GLP-1 is one of the primary factors contributing to host metabolic functions[26]. Additionally, interestingly, we identified HYA as a low-affinity ligand that activated EP3 in the gut, thereby promoting intestinal peristalsis. In actuality, low-affinity ligands cannot activate receptors in peripheral tissues and the intestine under normal conditions because of their low concentrations. However, under postprandial conditions, low-affinity ligands might activate local receptors in the gut based on temporary increases in concentrations of these nutrient-derived metabolites within the gut. Most GPCRs activated by lipid ligands display low-affinity binding with their many ligands[30]. Therefore, further study is warranted to clarify this transient nutrient-stimulated mechanism associated with nutrient-derived metabolites via low-affinity ligand–GPCR interactions in the intestine.

Here, we demonstrated that gut microbiota exerted protective effects against HFD-induced obesity and improved glucose homeostasis associated with several mechanisms involving gut microbial metabolites of dietary PUFAs: (1) gut microbiota converted LA to HYA, reducing the adipose inflammation otherwise induced by excessive dietary LA via the AA cascade; (2) HYA activated GPR40 and GPR120 and promoted GLP-1 secretion; and (3) HYA suppressed lipid absorption by promoting intestinal peristalsis via EP3. These findings not only potentially represent a central mechanism underlying interplay between commensal bacteria and the host for energy homeostasis via dietary lipids but also contribute to the development of functional foods for the prevention of metabolic disorders, such as obesity and type 2 diabetes mellitus, through tailoring the use of HYA. In addition, the identification of metabolites exhibiting high levels of bioactivity in other low-concentration gut microbial PUFA metabolites might promote developments of potential therapeutic relevance for the treatment of several diseases.

## Methods

**FA analysis**. The structures of the gut microbial metabolites of LA and αLA used in this study (Supplementary Fig. 1) were synthesized according to previously published methods (purity ≥ 98%)[17]. In brief, the reaction mixtures were prepared in each vial that contained 0.1% (wt/vol) FA complexed with BSA [0.02% (wt/vol)] in 20 mM KPB (pH 6.5) and purified enzymes (CLA-HY, CLA-DH, CLA-DC, and CLA-ER) in various combinations. The mixtures were added with 5 mM NADH, 5 mM NAD +, 0.1 mM FMN, or 0.1 mM FAD, and were gently shaken (120 strokes per minute) at 37 °C for 12 h under microaerobic conditions. Liquid chromatography tandem mass spectrometry (LC-MS/MS)-based lipidomics was performed, as described previously[17,31]. Briefly, samples were subjected to solid-phase extraction using a Sep-Pak C18 cartridge (Waters, Tokyo, Japan) with a deuterium-labeled internal standard (arachidonic acid-d8, leukotriene B4-d4, 15-hydroxyeicosatetraenoic acid-d8, prostaglandin E2-d4). Lipidomic analyses were performed using a high-performance LC system (Waters, UPLC) with a linear ion-trap quadrupole mass spectrometer (QTRAP 5500; AB SCIEX, Framingham, MA, USA) equipped with an Acquity UPLC BEH C18 column (1.0 mm × 150 mm × 1.7 μm; Waters). MS/MS analyses were conducted in negative-ion mode, and FA metabolites were identified and quantified by multiple-reaction monitoring. Quantitation was performed using calibration curves constructed for each compound, and recoveries were monitored using added deuterated internal standards.

To quantify individual FAs, samples (~50 mg cecal content or 100 μL plasma) with an internal control (C19:0) were homogenized in methanol (1 ml), and added chloroform (2 ml) and 0.5 M potassium chloride (0.75 mL) to extracts lipids. The collected lipid layers were dried, and the samples were resuspended with chloroform:methanol (1:3, v/v) for LC-MS/MS analysis. FAs were analyzed using an Acquity UPLC system coupled to a Waters Xevo TQD mass spectrometry (Waters) and separated on an ACQUITY UPLC BEH C18 column (2.1 × 150 mm, 1.7 μm; Waters) using an acetonitrile–isopropanol (9:1; v/v) gradient in 0.1% acetic acid aqueous solution (55% acetonitrile–isopropanol for 5 min, 55–100% acetonitrile–isopropanol from 5–16 min, 100% acetonitrile–isopropanol to 18 min, and then back to the initial 55% acetonitrile–isopropanol concentration at 18.1 min for 1.9 min for column equilibration). The flow rate was 0.4 mL/min, and column temperature was maintained at 50 °C. MS detection was performed in negative ionization mode, and the source capillary voltage was set to 3000 V. The desolvation and source temperatures were set at 500 °C and 150 °C, respectively. Individually optimized multiple-reaction-monitoring parameters were determined

for target compounds using standards. Subsequently, gas chromatography-mass spectrometry (GC-MS) were employed to quantify specific metabolites; LA, 9c15c-18:2, 10t12c-18:2, CLA1, and CLA3 because of co-elution on the LC-MS/MS analysis. Extracted fatty acids were dissolved in (Trimethylsilyl) diazomethane (abt. 10%)-hexane solution (Wako), and incubated at room temperature for 1 h to undergo methylation. The samples were dried, and the residues were resuspended with chloroform:methanol (1:3, v/v) for GC-MS analysis. FA methyl esters were analyzed using GC-MS QO2010 Ultra (Shimadzu) and separated on a VF-WAXms column (0.25 mm × 30 m, 1 μm; Agilent technologies). The initial oven temperature was held at 100 °C; ramped to 240 °C at a rate of 70 °C/min; and held at 240 °C for 37 min. Helium was used as a carrier gas at a flow rate of 1 mL/min. The temperatures of the EI ion source and injector were 200 and 260 °C, respectively. The electron energy was 70 eV. Full-scan mass spectra were recorded in the 35–400 m/z range. Quantification was done by integration of the extracted ion chromatogram peaks for the specific ion species of individual FAs using a calibration curve with standard FA.

**Animals**. C57BL/6 J, *Gpr40*-deficient, and *Gpr120*-deficient mice were housed under a 12-h light–dark cycle and given NC (CE-2; CLEA, Tokyo, Japan). All animal experiments were performed using mice from more than 4 litters per group. All experimental procedures involving mice were performed according to protocols approved by the Committee on the Ethics of Animal Experiments of the Tokyo University of Agriculture and Technology (permit No. 28–87). *Gpr40*- and *Gpr120*-deficient mice were generated on a C57BL/6 J background (Supplementary Figs. 6 and 7).

For short-term treatment, 8-week-old C57BL/6 J mice were placed on an NC, a D12492 diet (60% kcal fat; Research Diets, New Brunswick, NJ, USA), or a modified D12492 diet for 2 weeks. For long-term treatment, 4-week-old C57BL/6 J were placed on a D12492 diet or a modified D12492 diet for 12 weeks. To avoid coprophagy, mice were housed in wire mesh metal mouse cages, and placed on a D12492 diet or a modified D12492 diet for 12 weeks. The compositions of the diets are provided in Supplementary Table 2.

Time-dependent profiling of PUFA metabolites in plasma, small intestine, and colon was performed following the oral administration of individual FAs (1 g/kg) dissolved in 0.5% carboxymethyl cellulose ammonium (CMC; Nacalai Tesque, Kyoto, Japan). Time-course effects of FA administration on GLP-1 and insulin secretion were analyzed.

GF ICR mice were housed in vinyl isolators under a 12-h light–dark cycle. For short-term treatment, 8-week-old GF mice were placed on an NC (50 kGy irradiated; CLEA) or D12492 diet (50 kGy irradiated; Research Diets) for 2 weeks. For generation of gnotobiotic mice, ICR mice were administered with the cocktail of lactic acid bacteria (as shown in Supplementary Fig. 9a) in drinking water at 3, 8, and 12 weeks of age (3 × 10⁸ CFU/mL, respectively). Then, 4-week-old gnotobiotic, and GF ICR mice were placed on a D12492 diet (50 kGy irradiated, Research Diets) for 12 weeks. All experimental procedures involving GF mice were performed according to protocols approved by the Institutional Animal Care and Use Committee of Keio University School of Medicine [permission No. 09036-(12)]. Mice were killed by anesthesia with somnopentyl, and all efforts were made to minimize suffering.

For FMT experiments, fecal content was collected from HFD-fed or HYA-supplemented HFD-fed obese mice for 12 weeks. The fecal content was transferred into GF ICR mice aged 8 weeks (3 mice/pellet) three times weekly by gavage. After 1 week, OGTT was assessed.

**Gut microbiota composition**. Cecal DNA was extracted from frozen samples using the FastDNA SPIN kit for feces (MP Biomedicals, Santa Ana, CA, USA) according to manufacturer instructions. Bacteria (*L. salivarius* JCM1044, *L. johnsonii* JCM2012, *L. acidophilus* JCM1028, and *L. gasseri* JCM1131) was obtained from the Japan Collection of Microorganisms of RIKEN BRC (Tokyo, Japan) and used as a standard specifically for the DNA-based determination of intestinal lactic acid bacterial count. Bacterial DNA was isolated using the MonoFas bacterial genomic kit IV (GLC science, Tokyo, Japan) according to manufacturer instructions. Quantitative real-time PCR analysis was performed using SYBR Premix Ex Taq II (TaKaRa, Shiga, Japan) and the StepOnePlus real-time PCR system (Applied Biosystems, Foster City, CA, USA). Standard curves for quantification comprised a series of 10-fold serial dilutions in the range of 10⁸ to 10⁰ copies of target 16 S rRNA genes. Bacterial primer sequences were as follows: *Lactobacillus* genus, 5′-AGCAGTAGGGAATCTTCCA-3′ (forward) and 5′- CACCGCTACACATGGAG-3′ (reverse)[32]; *L. salivarius*, 5′-CGAAACTTTCTTACACCGAATGC-3′ (forward) and 5′-GTCCATTGTGGAAGATTCCC-3′ (reverse); *L. johnsonii*, 5′- CACTA-GACGCATGTCTAGAG-3′ (forward) and 5′-AGTCTCTCAACTCGGCTATG-3′ (reverse); *L. acidophilus*, 5′-GTCTGATGGAGCAACGCCGC-3′ (forward) and 5′-ACCTTGCGGTCGTACTCCCC-3′ (reverse); and *L. gasseri*, 5′-GCCACATTGG GACTGAGACA-3′ (forward) and 5′-TTGCTCCATCAGACTTGCGT-3′ (reverse). Partial 16 S rRNA gene sequences were amplified targeting the hyper-variable regions v4 using primers 515 F (5′-TCGTCGGCAGCGTCAGATGTGTA TAAGAGACAGGTGYCAGCMGCCGCGGTAA-3′) and 806 R (5′-GTCTCGTG GGCTCGGAGATGTGTATAAGAGACAGGGACTACHVGGGTWTCTAAT-3′). Amplicons generated from each sample were subsequently purified using AMPure XP (Beckman Coulter, Brea, CA, USA). The 16 S rRNA sequence data generated by

the MiSeq sequencer (Illumina, San Diego, CA, USA) using the MiSeq Reagent kit (version 3.0; 600 cycles) were processed by the quantitative insights into microbial ecology (QIIME 1.8.0; http://qiime.org/) pipeline. Data analysis was performed using MiSeq Reporter software with the SILVA database (Illumina). Diversity analyses were performed using the QIIME script *core_diversity_analyses.py*. The statistical significance of sample groupings was assessed using permutational multivariate analysis of variance (QIIME script *compare_categories.py*).

**Histology**. Adipose tissues were fixed in 10% neutral-buffered formalin, embedded in paraffin, and sectioned (7 μm). Hematoxylin and eosin staining was performed using standard techniques. To measure adipocyte area, the diameters of ≥100 cells from seven sections in each group were measured using ImageJ software (National Institutes of Health, Bethesda, MD, USA). The results were expressed as the average of >10 fields examined. Immunohistochemistry was performed using antibodies against F4/80 (rat; 1:1000; Abcam, ab6640, Cambridge, UK), Caveolin-1 (mouse; 1:200; BD Biosciences, 610406, Franklin Lakes, NJ, USA), and DAPI (1:5000; Roche, 10236276001, Basel, Switzerland), as previously described[33]. In brief, sections were fixed with 4% paraformaldehyde, washed in PBS, and treated with 5% BSA in PBS. The sections were permeabilized with 0.1% Triton X-100 (Sigma) and immunostained with each primary antibody. This was followed by signal development using secondary antibodies conjugated with a fluorescent marker.

**Biochemical analyses**. Blood glucose concentrations were measured using a One Touch Ultra (LifeScan, Milpitas, CA, USA). The concentrations of plasma and fecal triglycerides (LabAssay Triglyceride; Wako Pure Chemical Co. Ltd., Osaka, Japan), total cholesterol (LabAssay Cholesterol; Wako Pure Chemical Co. Ltd.), PYY (Mouse/Rat PYY ELISA Kit, Wako), GLP-1 (active) ELISA kit (Merck Millipore, Billerica, MA, USA), and an insulin ELISA Kit (Shibayagi, Gunma, Japan) were measured according to manufacturer instructions. For GLP-1 measurement, plasma samples and culture media were treated with a dipeptidyl peptidase IV (DPP-IV) inhibitor (Merck Millipore) to prevent degradation of active GLP-1.

For GTT, 24-h-fasted obese mice were given 1.5 mg of glucose/gram of body weight intraperitoneally (i.p.). For ITT, 3-h-fasted obese mice were administered human insulin (0.75 mU/g by i.p.; Sigma-Aldrich, St. Louis, MO, USA). Plasma glucose concentration was monitored before injection and at 15-, 30-, 60-, and 120-min post-injection.

**PGE2 measurement**. To assay total-adipose and small-intestinal PGE2 concentration, frozen tissues were homogenized in TNE buffer containing 10 mM Tris-HCl (pH 7.4), 150 mM NaCl, 1 mM EDTA, 1% Nonidet P-40, 50 mM NaF, 2 mM $Na_3VO_4$, 10 g/mL aprotinin, 1% phosphatase-inhibitor cocktail (Nacalai Tesque), and 100 μM indomethacin (Wako Pure Chemical Co. Ltd.). The homogenates were centrifuged at 3000 *g* and 4 °C for 20 min and stored at −80 °C until further analysis. Total protein levels in the supernatants were measured using a BCA protein assay kit (Thermo Fisher Scientific, Waltham, MA, USA). PGE2 concentrations in the supernatants were determined by ELISA kit (Cayman Chemical, Ann Arbor, MI, USA), standardized to the amount of total protein in supernatant, and presented as the amount of PGE2/mg of protein in supernatant.

**RNA isolation and quantitative reverse transcriptase PCR**. Total RNA was extracted using an RNeasy mini kit (Qiagen, Hilden, Germany) and RNAiso Plus reagent (TAKARA). cDNA was transcribed using RNA as templates and Moloney murine leukemia virus reverse transcriptase (Invitrogen, Carlsbad, Cam USA). quantitative reverse transcriptase (qRT)-PCR analysis was performed using SYBR Premix Ex Taq II (TAKARA) and the StepOnePlus real-time PCR system (Applied Biosystems, Foster City, CA, USA), as described previously[34,35]. In brief, the reaction was performed at 95 °C for 30 s, followed by 40 cycles of 95 °C for 5 s, 58 °C for 30 s and 72 °C for 1 min. The dissociation stage was analyzed at 95 °C for 15 s, followed by 1 cycle of 60 °C for 1 min and 95 °C for 15 s. Primer sequences were as follows: *Cla-hy*, 5′-TGGGGGCGTTATTTATGGT-3′ (forward) and 5′-GGCAA-CATACCCAGTCGTG-3′ (reverse); *Cla-dh*, 5′-TTTGGGAATTGCCGACTCGT-3′ (forward) and 5′-GAACAAAGCCAGGGCACATC-3′ (reverse); *Cla-er*, 5′-TTTGTCGTCGTGGATACCCC-3′ (forward) and 5′-CGGAA-CAATCAATCGTCGCA-3′ (reverse); *16S*, 5′-ACTCCTACGGGAGGCAGCAGT-3′ (forward) and 5′-ATTACCGCGGCTGCTGGC-3′ (reverse); *Tnfa*, 5′-GGCAGGTCTACTTTGGAGTC-3′ (forward) and 5′-TCGAGGCTCCAGT-GAATTCG-3′ (reverse); *F4/80*, 5′-GATGTGGAGGATGGGAGATG-3′ (forward) and 5′-ACAGCAGGAAGGTGGCTATG-3′ (reverse); *Mcp1*, 5′-AATCTGAAGC-TAATGCATCC-3′ (forward) and 5′-GTGTTGAATCTGGATTCACA-3′ (reverse); and *18 S*, 5′-ACGCTGAGCCAGTCAGTGTA-3′ (forward) and 5′-CTTA-GAGGGACAAGTGGCG-3′ (reverse).

**OGTT**. Following a 16-h fast, mice were administered LA or HYA (1 g/kg) dissolved in 0.5% CMC by gavage, and after 2 h, mice were administered glucose (5 g/kg) for OGTT[36]. Plasma glucose concentration was monitored before injection and at 15-, 30-, 60-, and 120-min post-injection.

**Cell culture**. STC-1 cells (murine enteroendocrine cells; ATCC, Manassas, VA, USA) were cultured in Dulbecco's modified Eagle medium (DMEM) containing 5% fetal bovine serum (FBS) and 15% horse serum and maintained at 37 °C and 5% $CO_2$. For measurement of GLP-1 secretion, cells were plated in 24-well plates (1 × $10^5$ cells/well) and cultured for 48 h before initiating GLP-1 secretion. Each well was treated with FA for 0.5 h to 1 h, and the supernatant was collected in the presence of a DPP-IV inhibitor. For siRNA knockdown, STC-1 cells were transfected with 30 nM siRNA (*Gpr40*: CAAUGUGGCUAGUUUCAUA; and *Gpr120*: CGAAAUGACUUGUCUGUUA; Dharmacon, Lafayette, CO, USA) using Lipofectamine 2000 transfection reagent (Invitrogen) according to manufacturer instructions. For inhibitor treatment, STC-1 cells were pretreated with the MEK inhibitor U0126 (10 μM; Wako), the PLC inhibitor U73122 (1 μM; Wako), or the CaMKII inhibitor KN-62 (10 μM, Wako) for 30 min prior to the addition of FAs.

GLUTag cells used in this study were generated by Dr. Drucker from the University of Toronto[37], cultured in DMEM containing 10% FBS, and maintained at 37 °C and 5% $CO_2$. For measurement of GLP-1 secretion, cells were plated in 24-well plates (1 × $10^5$ cells/well) and cultured for 48 h before initiating GLP-1 secretion. Each well was treated with FA for 0.5 h to 1 h, and supernatant was collected in the presence of a DPP-IV inhibitor.

Flp-In T-REx HEK293 cells were obtained from Invitrogen, and for generation of HEK293 cells expressing human GPR40 or GPR120, the HEK293 cells were transfected with a mixture of pcDNA5/FRT/TO- FLAG-mGPR40 or mGPR120 and pOG44, respectively, using the Lipofectamine reagent (Invitrogen)[38,39], and the cells were cultured in DMEM containing 10 μg/mL blasticidin S (Funakoshi, Tokyo, Japan), 100 μg/mL hygromycin B (Gibco, Grand Island, NY, USA), and 10% FBS. Next, the cells were incubated at 37 °C in an atmosphere of 5% $CO_2$ and further cultured under various conditions. For $[Ca^{2+}]i$-response analysis, cells were plated onto 96-well black plates (3 × $10^4$ cells/well) and incubated for 24 h, and each well was treated with or without doxycycline (10 μg/mL) for 24 h. After treatment, cells were further incubated in Hank's balanced salt solution (HBSS; pH 7.4) containing calcium assay kit component A (Molecular Devices, Sunnyvale, CA, USA) for 1 h at 37 °C. Each FA used in the Functional Drug Screening System (FlexStation 3 Multi-Mode Microplate Reader; Molecular Devices) assay was dissolved in HBSS and prepared in another set of 96-well plates, which were placed in the Functional Drug Screening System to monitor the mobilization of $[Ca^{2+}]i$.

Human EP3-expressing Chem-1 cells were measured for the $[Ca^{2+}]i$ response induced by each FA using a FLIPR TETRA system (Molecular Devices) with an ICCD camera (Eurofins Scientific, Brussels, Belgium).

**Intestinal peristalsis**. Wild-type mice were orally administered the EP3 agonist (0.1 mg/kg; Sigma-Aldrich) or each FA (1 g/kg) containing charcoal suspended in 0.5% CMC. After 2 h, the mice were killed, and the gastrointestinal tracts were harvested. Intestinal peristalsis was expressed as the distance of the charcoal from the cecum to the start of the charcoal as measured with a ruler.

**Murine intestinal motility**. Intestinal motility in response to each FA and EP3 agonist was determined according to a previously described protocol[40]. In brief, segments of the distal ileum were removed, cut along the mesenteric border, and pinned flat to the bottom of a dish coated with silicone rubber and filled with cold Krebs–Ringer solution. The tissue was cut parallel to the longitudinal smooth muscle direction. The preparations were placed in organ baths filled with 5 mL Krebs–Ringer gassed with 95%$O_2$/5%$CO_2$ at 37 °C and connected to isometric force transducers (MLT0420; ADInstruments, BellaVista, Australia) by surgical sutures. A four-channel bridge amplifier (FE224; ADInstruments) and a PowerLab 4/26 (ADInstruments) were used to record the tension and amplitudes of spontaneous contractions. During a 1-h equilibration, basal tensions of all preparations were adjusted by 1 mN to 2 mN, and tetrodotoxin (10 μM) and piroxicam (0.1 μM) were added in order to remove neural activity and basal prostaglandin production, respectively, 30 min before experiments.

**Bacterial culture**. *Lactobacillus* strains (Supplementary Table 1) were obtained from the Japan Collection of Microorganisms (Saitama, Japan). Each strain was inoculated into 15 mL of MRS medium in screw-capped tubes at 37 °C and then shaken at 120 strokes per minutes for 1 to 2 days. After cultivation, cells were harvested by centrifugation (1,500 × *g* for 10 min) and washed twice with 0.85% NaCl. The reactions were performed in test tubes containing 1 mL of reaction mixture [washed cells, 100 mM potassium phosphate buffer (pH 6.5), and 1 mg/mL LA] under anaerobic conditions using an Aneropack "Kenki" (Mitsubishi Gas Chemical Co., Ltd., Tokyo, Japan) in a sealed chamber and then shaken at 120 strokes per minutes and 37 °C for 24 h. For detection of LA, HYA, and HYB in the reaction mixture, lipids were extracted with 5 mL of chloroform/methanol/1.5% KCl in $H_2O$ (2:2:1, by volume), according to the Bligh–Dyer procedure and concentrated by evaporation under reduced pressure. The resulting lipids were analyzed by gas chromatography using a Shimadzu (Kyoto, Japan) GC-1700 gas chromatograph equipped with a flame ionization detector and a split injection system and fitted with a capillary column (SPB-1, 30 m × 0.25 mm i.d.; Supelco; Sigma-Aldrich).

**Statistical analysis**. All values are presented as the mean ± standard error of the mean. Differences between groups were examined for statistical significance using Student's *t* test or the Tukey–Kramer post hoc test (≥3 groups). A *P* < 0.05 was considered statistically significant. False discovery rates (*q*-value) of metabolomic and 16 S rRNA gene sequencing data were estimated using the Benjamini–Hochberg procedure.

**Reporting summary**. Further information on research design is available in the Nature Research Reporting Summary linked to this article.

## Data availability

A reporting summary for this article is available as a Supplementary Information file. The source data underlying Figs. 1–7, Supplementary Figs. 1–9, Supplementary Table 1, and lipid metabolome, are provided as a Source Data file. 16S rRNA sequence data have been deposited into the DNA Data Bank of Japan (DDBJ) under the accession no. DRA008263 and no. DRA008264. All other data generated or analyzed during this study are included in this published article and its Supplementary Information files or are available from the corresponding authors upon reasonable request.

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

## Acknowledgements

This work was supported by research grants from the JSPS and MEXT KAKENHI (grant nos. (JP15H05344 and JP16H01355, respectively) to I.K., AMED (no. JP18gm1010007) to I.K., CIHR Foundation (grant 154321) to D.J.D., the Institute for Fermentation Osaka to IK, and the Smoking Research Foundation to IK. We thank NITTO PHARMACEUTICAL INDUSTRIES, LTD. for supplying purified HYA, Kanako Igarashi, Miwa Yoshimoto, and Mie Honda for their help with metabolome analysis, Kumiko Tachikawa, Mai Arita, and Shizuka Kasuga for their help with fatty acid analysis and in vitro assays, Yukie Katayama for 16S rRNA gene sequence analysis, Kohey Kitao and Keiko Hisa for their valuable discussions, and Dr. Tadahiro Kitamura for the GLUTag cells.

## Author contributions

J.M. performed the experiments and wrote the paper; M.I. performed the experiments, interpreted data, and wrote the paper; K.W. performed the experiments and interpreted data; Shin.K. performed the experiments and interpreted data; H.M. performed the experiments; Shig.K. performed the experiments and interpreted the data; X.L. performed

the experiments; A.I. interpreted data; J.I. interpreted the data; Y.S. interpreted the data; Te.M. performed the experiments; T.S. interpreted the data; J.O. interpreted the data; Ta.M. interpreted the data; D.J.D. interpreted the data; M.A. performed the experiments and interpreted the data; H.I. interpreted the data; I.K. supervised the project, interpreted the data, and wrote the paper; I.K. had primary responsibility for the final content. All authors read and approved the final manuscript.

## Additional information

**Competing interests:** The authors declare no competing interests.

