## [Peer Review File · Nature Communications]

REVIEWERS' COMMENTS:

Reviewer #1 (Remarks to the Author):

In general, I find the argument for beneficial metabolic effects of microbiome-generated PUFA metabolites to be convincing. However, I continue to believe that there is no evidence for a direct role of GLP-1 in these effects, as follows:

1. It is clear that PUFA metabolites do indeed stimulate the release of GLP-1, both in vitro and in vivo, through GPR40 and 120. However, as noted previously, the fact that the glycemic and insulin responses in the IPGTT are identical to the changes observed in the OGTT argues strongly that there is no role for the incretin hormones in this effect, either acutely or chronically.

2. The findings that GPR40 and 120 KO abrogates HYA-induced changes do not argue for a role of GLP-1 in this phenomenon, as these receptors are both expressed by other cell types, most notably the beta cell. Hence, effects of HYA on the beta cell, which have not been examined in the present study, could explain the identical IPGTT and OGTT results independent of any role for incretin hormones. Further, the argument that plasma levels of HYA are lower than those of LA does not preclude an effect on the beta cell, as their in vivo potencies on the beta cell (relative to their plasma concentrations) have not been explored.

3. Without the GLP-1R KO mice, which have been appropriately removed (as previously discussed) and in the absence of studies utilizing a GLP-1R antagonist, there remains no evidence to demonstrate that any effects, either metabolic or weight related, are due to changes in GLP-1 secretion. Thus, throughout the manuscript, comments such as "via", "due to", "directly", "thereby influencing", "GLP-1-mediated" and "through" GLP-1 should be modified to indicate at best "an association with" changes in GLP-1.

4. The authors are thanked for normalizing the in vitro secretion data, as requested.

Response to the reviewer's comments:

To Reviewer #1:

In general, I find the argument for beneficial metabolic effects of microbiome-generated PUFA metabolites to be convincing. However, I continue to believe that there is no evidence for a direct role of GLP-1 in these effects, as follows:

We appreciate the reviewer's thoughtful comments and constructive suggestions regarding our work again. We totally agree with the reviewer's comments that PUFA metabolites have host metabolic benefits beside promotion of GLP-1 release and that there is no evidence for a direct role of GLP-1 in these effects. Therefore, we have further revised the manuscript according to the reviewer's suggestion. Our point-by-point responses are provided below.

Q1.

It is clear that PUFA metabolites do indeed stimulate the release of GLP-1, both in vitro and in vivo, through GPR40 and 120. However, as noted previously, the fact that the glycemic and insulin responses in the IPGTT are identical to the changes observed in the OGTT argues strongly that there is no role for the incretin hormones in this effect, either acutely or chronically.

Response:

We appreciate the reviewer's valuable comments. According to the reviewer's comments, we have explained the effects of insulin response and GLP-1 release by acute HYA administration separately throughout the revised manuscript.

Q2.

The findings that GPR40 and 120 KO abrogates HYA-induced changes do not argue for a role of GLP-1 in this phenomenon, as these receptors are both expressed by other cell types, most notably the beta cell. Hence, effects of HYA on the beta cell, which have not been examined in the present study, could explain the identical IPGTT and OGTT results independent of any role for incretin hormones. Further, the argument that plasma levels of HYA are lower than those of LA does not preclude an effect on the beta cell, as their in vivo potencies on the beta cell (relative to their plasma concentrations) have not been explored.

Response:

We appreciate the reviewer's comments. As the reviewer mentioned, since GPR40 is abundantly expressed in pancreatic beta-cells, we also agree that HYA-mediated effects may induce insulin response via other mechanisms besides GLP-1 release. Therefore, we have indicated the possibility of GLP-1-independent insulin action by gut microbial PUFA metabolites in the Discussion section. Additionally, we have indicated the possibility of circulating HYA-mediated effects on insulin response in the Discussion section.

Q3.

Without the GLP-1R KO mice, which have been appropriately removed (as previously discussed) and in the absence of studies utilizing a GLP-1R antagonist, there remains no evidence to demonstrate that any effects, either metabolic or weight related, are due to changes in GLP-1 secretion. Thus, throughout the manuscript, comments such as "via", "due to", "directly", "thereby influencing", "GLP-1-mediated" and "through" GLP-1 should be modified to indicate at best "an association with" changes in GLP-1.

Response:

We appreciate the reviewer's constructive suggestions. We have modified "via", "due to", "directly", "thereby influencing", "GLP-1-mediated", and "through" to "an association with" throughout the revised manuscript.

Q4.

The authors are thanked for normalizing the in vitro secretion data, as requested.

Again, we sincerely thank the reviewer for the time and effort invested in improving our manuscript.